# On the Prognosis of Multifocal Glioblastoma: An Evaluation Incorporating Volumetric MRI

Johannes Kasper [1,*,†], Nicole Hilbert [1,†], Tim Wende [1], Michael Karl Fehrenbach [1], Florian Wilhelmy [1], Katja Jähne [1], Clara Frydrychowicz [2], Gordian Hamerla [3], Jürgen Meixensberger [1] and Felix Arlt [1]

1   Department of Neurosurgery, University Hospital Leipzig, Liebigstraße 20, 04103 Leipzig, Germany; nicolehilbert@t-online.de (N.H.); tim.wende@medizin.uni-leipzig.de (T.W.); michael.fehrenbach@medizin.uni-leipzig.de (M.K.F.); florian.wilhelmy@medizin.uni-leipzig.de (F.W.); katja.jaehne@medizin.uni-leipzig.de (K.J.); juergen.meixensberger@medizin.uni-leipzig.de (J.M.); felix.arlt@medizin.uni-leipzig.de (F.A.)
2   Division of Neuropathology, University Hospital Leipzig, Liebigstraße 26, 04103 Leipzig, Germany; Clara.Frydrychowicz@medizin.uni-leipzig.de
3   Institute of Neuroradiology, University Hospital Leipzig, Liebigstraße 20, 04103 Leipzig, Germany; gordian.hamerla@medizin.uni-leipzig.de
*   Correspondence: johannes.kasper@medizin.uni-leipzig.de; Tel.: +49-341-9717892; Fax: +49-341-9717509
†   Contributed equally.

**Abstract:** Primary glioblastoma (GBM), IDH-wildtype, especially with multifocal appearance/growth (mGBM), is associated with very poor prognosis. Several clinical parameters have been identified to provide prognostic value in both unifocal GBM (uGBM) and mGBM, but information about the influence of radiological parameters on survival for mGBM cohorts is scarce. This study evaluated the prognostic value of several volumetric parameters derived from magnetic resonance imaging (MRI). Data from the Department of Neurosurgery, Leipzig University Hospital, were retrospectively analyzed. Patients treated between 2014 and 2019, aged older than 18 years and with adequate peri-operative MRI were included. Volumetric assessment was performed manually. One hundred and eighty-three patients were included. Survival of patients with mGBM was significantly shorter ($p < 0.0001$). Univariate analysis revealed extent of resection, adjuvant therapy regimen, residual tumor volume, tumor necrosis volume and ratio of tumor necrosis to initial volume as statistically significant for overall survival. In multivariate Cox regression, however, only EOR (for uGBM and the entire cohort) and adjuvant therapy were independently significant for survival. Decreased ratio of tumor necrosis to initial tumor volume and extent of resection were associated with prolonged survival in mGBM but failed to achieve statistical significance in multivariate analysis.

**Keywords:** glioblastoma; volumetric assessment; multifocal glioblastoma

## 1. Introduction

Isocitrate dehydrogenase wildtype glioblastoma (GBM) is the most common brain-derived malignancy; it averagely affects patients older than 60 years and is more often diagnosed in men [1]. Characterized by high mitotic activity, aggressive invasive behavior, central necrosis and neo-angiogenesis, it is classified as WHO grade IV [2]. Standard therapy consists of maximum safe resection followed by adjuvant concomitant radio-chemotherapy and maintenance chemotherapy, both applying temozolomide as standard of care [3–5]. Despite intensive research efforts, prognosis remains poor with one-year survival of about 40% [1], and several mechanisms responsible for therapeutic resistance, including the blood-brain-barrier, intra-tumoral heterogeneity or stem-like cells, have been identified [6]. Here, tumor treating fields [7] represent a new treatment approach. Several clinical parameters have been identified as relevant to predict overall survival such as O6-methylguanine DNA methyltransferase promoter methylation (MGMT) [8], age at date of diagnosis [9] or clinical performance [10]. Radiological parameters, gathered

from peri-operative magnetic resonance imaging (MRI), also proved to be of prognostic value, including residual tumor volume, extent of resection, volume of necrosis or unifocal vs. multifocal GBM [11–14]. Especially clinical management of multifocal GBM (mGBM) is often difficult due to reduced rates of gross total resection (or craniotomy in general) causing significantly shorter overall survival (OS) [15]. While clinical factors and molecular characteristics have been investigated for both unifocal glioblastoma (uGBM) and mGBM [16–19], data about the prognostic impact of radiological parameters for multifocal GBM are scarce [20,21]. Hence, this retrospective study investigated whether additional MRI characteristics (initial tumor volume, necrosis volume, residual volume) of mGBM provide prognostic information and how these data statistically implement into overall prognosis when established clinical and molecular factors are considered.

## 2. Methods

### 2.1. Patient Selection and Treatment

Data collection and analysis were approved by the ethical committee of Leipzig Medical Faculty (336/20-ek) and carried out in accordance with valid data protection guidelines at University Hospital Leipzig. Informed consent for retrospective data analysis was obtained from all patients. The medical records were checked for all patients with first diagnosis of IDH-wildtype glioblastoma between 1 January 2014 and 31 December 2019 treated at Leipzig University Hospital. Patients aged less than 18 years or without cranial MRI before and within 72 h after surgery were excluded. All tumor cases in our department and certified neurooncological center were discussed in a weekly, interdisciplinary tumor board, and therapy regimen was determined based on the current treatment guidelines for glioma therapy.

### 2.2. Clinical, Pathological and Radiological Assessment

Medical records were analyzed for age at date of diagnosis, post-operative clinical performance, gender and adjuvant therapy regimen. The date of diagnosis was set as the date of surgery with neuropathological proof of glioblastoma. The medical research council neurological performance scale (MRC-NPS) adjusted by Bleehen et al. [22] was used to assess neurological performance with (1) no neurological deficit, (2) some neurological deficit but function adequate for useful work, (3) neurological deficit causing moderate functional impairment (difficulty to move limbs, moderate dysphasia, moderate paresis, some visual disturbance), (4) neurological deficit causing major functional impairment (inability to move limbs, gross speech or visual disturbances) and (5) inability to make conscious responses.

Histopathological diagnosis and immunohistochemical status were extracted from neuropathology reports. IDH mutation status and MGMT promoter methylation of all GBM samples were determined using immunohistochemistry and pyrosequencing or nucleic acid amplification followed by pyrosequencing.

Overall survival (OS) was defined as the time between date of neurosurgery and date of death. The date of death, if not provided by our hospital database, was collected from the Leipzig Cancer Registry. Dates were assessed on 31 August 2020. If death did not occur by then or if patients were lost to follow-up, the date of last contact to our department was integrated into statistical analysis as censored value.

Pre-operative necrosis volume (CEV-, T1-hypointense tumor volume surrounded by contrast-enhancing tumor volume), initial tumor volume (IV, sum of contrast-enhancing tumor volume and CEV-), residual tumor volume (RV, contrast-enhancing tumor volume assessed from subtraction sequences of T1 MR imaging with and without contrast) and extent of resection (EOR, ratio of RV and IV in percent) were retrospectively determined revising MRI T1 sequences with and without contrast. Volumetric assessment was manually carried out employing iPlan Cranial software (version 3.0.5, Brainlab AG, Munich, Germany). If burr hole trepanation with needle biopsy was performed, EOR was set as

0%. Multifocality was defined as separate (distance greater than 1 cm) contrast-enhancing lesions, independently from FLAIR hyperintensity.

### 2.3. Statistical Analysis

Statistical analysis was carried out for the entire cohort and sub-cohorts (unifocal vs. multifocal GBM) using SPSS statistics software version 24.0.0.2 (IBM, Armonk, NY, USA). First, assessed parameters were applied as continuous variables and analyzed via univariate Cox regression. Time-dependent receiver operator characteristic (ROC) analysis was then performed for statistically significant radiological parameters. When area under the curve (AUC) from ROC analysis surpassed 0.6, the optimal cutoff point was defined as the value that maximizes the Youden index (parameter value for which sensitivity + specificity − 1 is maximal). After radiological parameters were categorized according to cutoff values, a second univariate analysis was carried out with Cox regression and Kaplan–Meier estimate. Finally, all continuous variables with p values below 0.2 in univariate Cox regression were utilized for a multivariate analysis via proportional hazard Cox regression in order to investigate independent statistical relevance. Nonparametric variables were compared with Mann–Whitney *U* test.

Survival rates from Kaplan–Meier analyses are given with standard deviation. P-values below 0.05 were considered statistically significant. Hazard ratios (HRs) are provided with 95% confidence intervals (CIs) and were considered statistically significant if 1 was excluded by CI.

## 3. Results

### 3.1. Patient Cohort

Within the study period, 225 patients were newly diagnosed with primary GBM of which 42 were excluded (40 due to inadequate peri-operative MRI-imaging and 2 due to age under 18 years). One hundred and eighty-three patients (129 with unifocal and 54 with multifocal GBM) were eligible for this study. Baseline data are given in Table 1. Concerning age and gender ratio, the entire cohort is comparable to larger studies [1]. Statistically relevant differences between sub-cohorts were found for residual tumor volume, extent of resection and adjuvant therapy regimen. Patient survival of sub-cohorts is shown in Figure 1. OS was significantly shorter for patients suffering from mGBM (one-year survival $64.9 \pm 4.8\%$ (uGBM) vs. $16.9 \pm 6.4\%$ (mGBM), $p < 0.0001$).

**Table 1.** Baseline data. Decimal values represent averages with standard deviation.

|  |  | **All** | **Unifocal** | **Multifocal** | *p*-**Value** |
|---|---|---|---|---|---|
| Number of patients |  | 183 | 129 | 54 | - |
| Age (years) |  | $67.4 \pm 10.6$ | $66.9 \pm 10.9$ | $68.5 \pm 9.9$ | 0.24 |
| Gender ratio (male to female) |  | 1:0.48 | 1:0.5 | 1:0.45 | 0.68 |
| IV (mL) |  | $33.1 \pm 27$ | $32.4 \pm 25.6$ | $34.8 \pm 28.8$ | 0.44 |
| CEV- (mL) |  | $10.3 \pm 13.3$ | $11.3 \pm 14.0$ | $7.9 \pm 11.2$ | 0.34 |
| CEV-/IV |  | $0.25 \pm 0.2$ | $0.27 \pm 0.18$ | $0.19 \pm 0.16$ | 0.05 |
| RV (mL) |  | $9.6 \pm 16.3$ | $5.7 \pm 8.9$ | $20.5 \pm 24.8$ | **<0.0001** |
| EOR (%) |  | $67.6 \pm 37.4$ | $79.3 \pm 30.2$ | $36.9 \pm 36.4$ | **<0.0001** |
| MRC-NPS |  | $2.6 \pm 0.9$ | $2.6 \pm 0.9$ | $2.6 \pm 0.9$ | 0.29 |
| MGMT (%) |  | $22.3 \pm 20.6$ | $22.2 \pm 20.9$ | $22.5 \pm 20.1$ | 0.47 |
| Adjuvant therapy | w/o | 35 | 16 | 19 | |
|  | Rx | 34 | 26 | 8 | **0.02** |
|  | RCx | 114 | 87 | 27 | |

Statistical analysis was performed with Mann–Whitney U test. Bold *p*-values indicate statistical significance. CEV-: non-enhancing tumor area; CEV-/IC: ratio of CEV- to IV; EOR: extent of resection; IV: initial tumor volume; RV: residual tumor volume; MGMT: O6-methylguanine-DNA methyltransferase promoter methylation; MRC-NPS: medical research council neurological performance scale; RCx: radio-chemotherapy; Rx: radiotherapy.

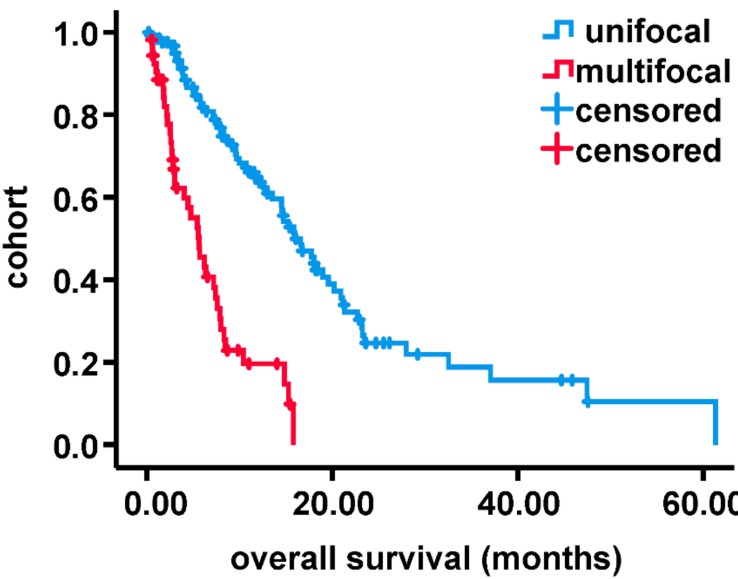

**Figure 1.** Compared survival of multifocal with unifocal glioblastoma by Kaplan–Meier analysis.

### 3.2. Analysis of MRI-Volumetric Parameters

Univariate analysis with continuous variables revealed statistical significance for RV (entire cohort, uGBM), EOR (entire cohort, uGBM and mGBM) and ratio of CEV- to IV (mGBM). Data are shown in Supplementary Table S1. Cutoff values (Supplementary Table S2) for those factors were calculated as stated within methods. AUCs were sufficient for statistical analysis. Optimal thresholds were 91.0% (uGBM), 82.5% (entire cohort) and 27.7% (mGBM) for EOR, 1.04 mL (entire cohort) and 0.41 mL (uGBM) for RV and 0.23 for ratio of CEV- to IV (CEV-/IV) in mGBM. Parameters were categorized according to cutoff values, and univariate analysis was carried out again (Supplementary Tables S3 and S4). Corresponding survival curves are presented in Figures 2 and 3. Please note that due to reduced prognosis, Kaplan–Meier analysis for mGBM is provided as six-month survival rates. Extent of resection above cutoff was significantly associated with prolonged survival in all cohorts (entire cohort: HR 0.36, one-year survival increased by 2-fold; uGBM: HR 0.56, one-year survival increased by 1.5-fold; mGBM: HR 0.47, six-months survival increased by 3-fold). CEV-/IV above the threshold of 0.23, on the other hand, was negatively associated with overall survival in mGBM (HR 2.2, six-month survival decreased by 2-fold). Reduced residual tumor volume increased survival significantly in the entire cohort (HR 0.46, one-year survival increased by 2-fold) but failed to show statistical significance in patients with uGBM.

### 3.3. Multivariate Analysis

Univariate Cox regression of other parameters (Supplementary Table S5) revealed adjuvant therapy regimen as significantly associated with overall survival in all three cohorts (all $p < 0.0001$), age and MGMT promoter methylation for the entire cohort and uGBM sub-cohort (all $p < 0.01$) and post-operative MRC-NPS for the entire cohort only ($p = 0.02$). Multivariate Cox regression was then performed utilizing all factors with $p$ values below 0.2 from univariate analysis and is presented in Table 2. Application of radio-chemotherapy prolonged OS in all cohorts and was the only parameter with independent statistical significance in mGBM (HR 0.429; $p = 0.002$). Extent of resection was significantly associated with overall survival in the entire cohort (HR 0.992; $p = 0.017$) and uGBM (H 0.997; $p = 0.003$). Other factors, especially RV, CEV- and CEV-/IV, were influential but failed to achieve statistical significance.

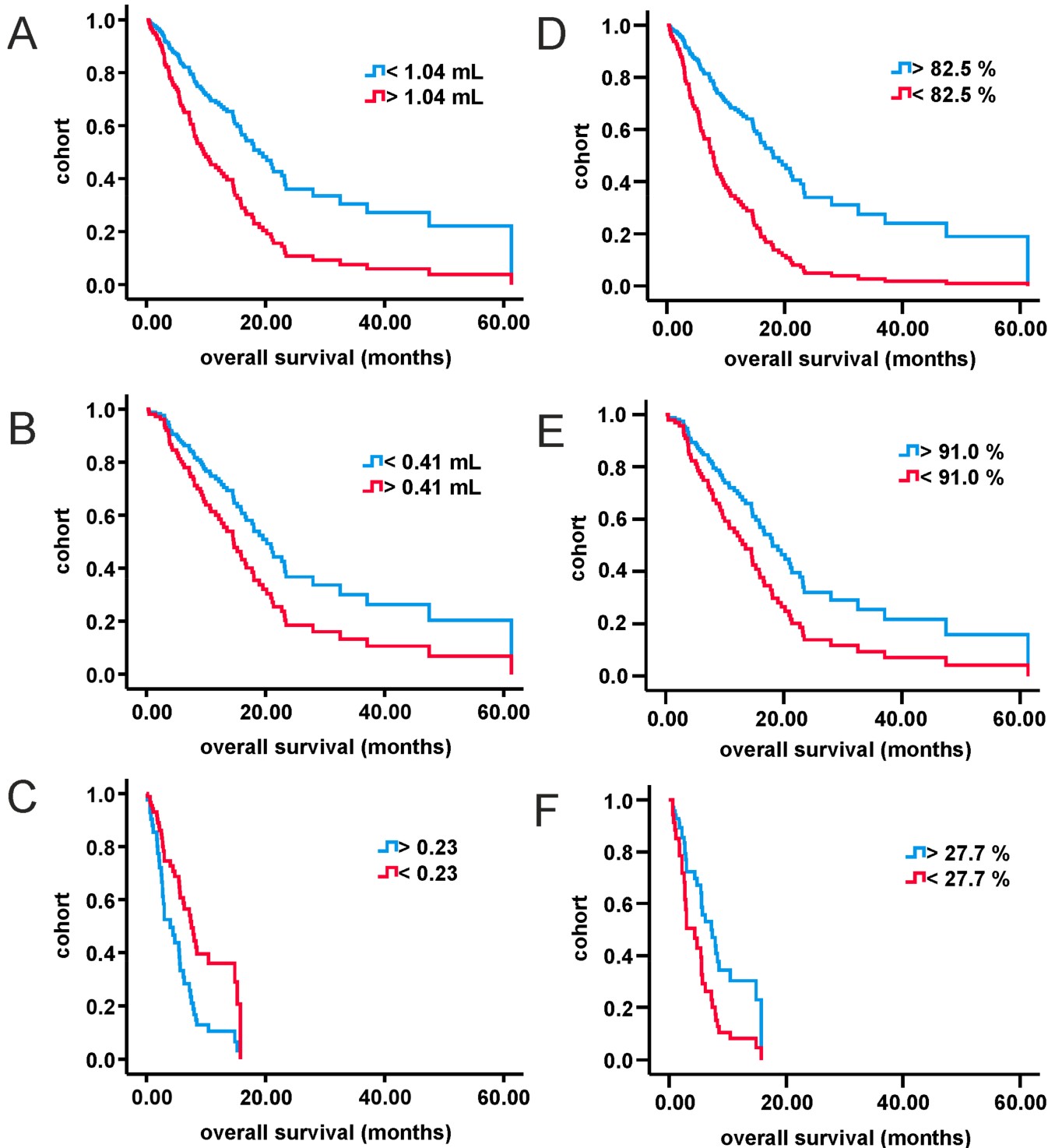

**Figure 2.** Cox regression of categorized volumetric parameters for all three study cohorts. (**A**): entire cohort, residual tumor volume; (**B**): uGBM, residual tumor volume; (**C**): mGBM, ratio of tumor necrosis volume to initial tumor volume; (**D**): entire cohort, extent of resection; (**E**): uGBM, extent of resection; (**F**): mGBM, extent of resection. mGBM: multifocal glioblastoma; uGBM: unifocal glioblastoma.

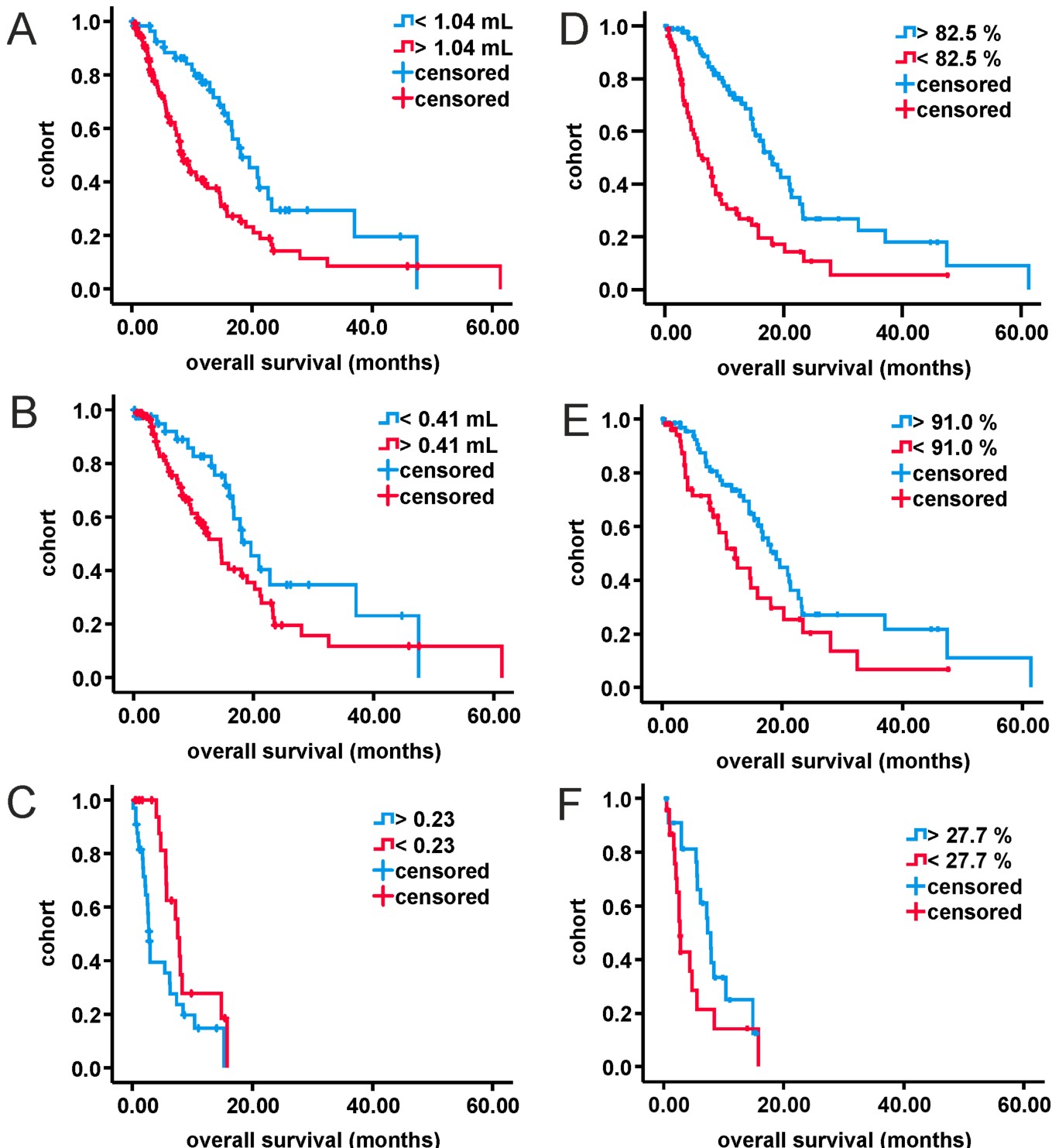

**Figure 3.** Kaplan–Meier analysis of categorized volumetric parameters for all three study cohorts. (**A**): entire cohort, residual tumor volume; (**B**): uGBM, residual tumor volume; (**C**): mGBM, ratio of tumor necrosis volume to initial tumor volume; (**D**): entire cohort, extent of resection; (**E**): uGBM, extent of resection; (**F**): mGBM, extent of resection. mGBM: multifocal glioblastoma; uGBM: unifocal glioblastoma.

**Table 2.** Multivariate Cox regression.

| | All | | | Unifocal | | | Multifocal | | |
|---|---|---|---|---|---|---|---|---|---|
| | HR | CI | *p*-Value | HR | CI | *p*-Value | HR | CI | *p*-Value |
| Age | 1.012 | 0.99–1.03 | 0.282 | 0.998 | 0.99–1.01 | 0.704 | - | - | - |
| MGMT | 0.989 | 0.98–1.0 | 0.086 | 1.026 | 1.0–1.06 | 0.07 | - | - | - |
| MRC-NPS | 0.924 | 0.7–1.2 | 0.58 | 1.055 | 0.74–1.49 | 0.759 | - | - | - |
| Adjuvant therapy | 0.352 | 0.24–0.51 | **<0.0001** | 0.391 | 0.23–0.67 | **0.001** | 0.429 | 0.27–0.7 | **0.002** |
| CEV- | - | - | - | - | - | - | 1.011 | 0.95–1.07 | 0.709 |
| CEV-/IV | 0.998 | 0.99–1.01 | 0.703 | - | - | - | 0.965 | 0.93–1.01 | 0.104 |
| RV | 1.011 | 1.0–1.03 | 0.175 | 1.037 | 1.0–1.08 | 0.085 | - | - | - |
| EOR | 0.992 | 0.985–0.998 | 0.017 | 0.977 | 0.962–0.99 | **0.003** | 0.998 | 0.99–1.01 | 0.699 |

Bold values indicate statistical significance. CEV-: non-enhancing tumor area; CEV-/IV: ratio of CEV- to IV; CI: 95% confidence interval; HR: hazard ratio; RV: residual tumor volume; EOR: extent of resection; IV: initial tumor volume; MGMT: O6-methylguanine-DNA methyltransferase promoter methylation; MRC-NPS: medical research council neurological performance scale.

## 4. Discussion

The presented study investigated the influence of MRI-volumetric parameters on overall survival with special emphasis on patients suffering from multifocal GBM. Although multivariate Cox regression revealed only adjuvant therapy regimen to be of independent significance, it is the first analysis to identify the ratio of tumor necrosis volume and initial tumor volume to be associated with prognosis and to provide an EOR threshold benefitting survival in patients suffering from mGBM.

mGBM represents a special sub-group of glioblastomas, accounting for around 10 to 20% of all cases and associated with significantly reduced overall survival [14,18,19,23,24] as was also shown for our cohort. Several, mostly retrospective, studies have investigated the prognostic effects comparing craniotomy (including gross total resection, GTR, and sub-total resection, STR) to biopsy only for patients with first diagnosis of multifocal glioma. Evidence was clearly in favor of craniotomy and GTR [25–27]. A study by Hassaneen et al. even showed comparable survival rates of mGBM patients to those with uGBM when multiple craniotomies were performed in one session to achieve GTR [28]. Di et al. [15], on the other hand, found out that resecting more than 85% of the initial tumor mass already improved overall survival in their cohort. The largest investigation so far by Haque et al. [16], relying on the US National Cancer Database and analyzing more than 40,000 cases (around 8000 with mGBM), not only found significant differences in surgery modalities when mGBM was compared to uGBM but also proved craniotomy superior to biopsy only in mGBM. Here, GTR was associated with greatest benefit for survival. Our data are congruent to those findings. Moreover, cutoff analysis revealed that already resecting more than 27.7% of initial tumor volume increased survival probability and six-month survival, respectively, underlining the importance of cytoreductive therapy via (partial) extirpation of macroscopically visible tumor tissue.

Concerning additional radiological parameters, tumor necrosis as one key feature of glioblastoma is of special interest. Large tumor necrosis volume was shown to be associated with reduced overall survival of GBM patients [29–31]. Moreover, post-hoc evaluation of the EORTC 26101 cohort revealed newly formed or progressive tumor necrosis during adjuvant chemotherapy (lomustin combined with or without bevacizumab) as a negative prognostic factor for patients with progressive glioblastoma [32]. Necrotic patterns of glioblastoma categorized via fractal analysis of pre-operative MRI also seem to be related to survival [33]. Recently, two publications by Henker et al. not only presented tumor to necrosis ratio to influence patient survival [34] but also to be associated with reduced frequency of seizures as onset symptom in glioblastoma patients [35]. Generally, tumor cell necrosis in GBM is initiated by acute ATP depletion via reduced oxidative phosphorylation resulting in the inability to maintain cross-membrane electrolyte and water homeostasis [36]. While several pathways might be included in promoting tumor cell necrosis and extracellular glutamate seem to play a pivotal role [37], the exact mechanisms

of necrosis in GBM remain unclear. Hence, explanations for the reverse connection between necrosis and prognosis in GBM are theoretical. There is evidence that hypoxia-induced necrosis promotes GBM cell migration and invasiveness, respectively, [38] most likely driven by increased levels of hypoxia-induced factor 1 which itself is well established as a driver of cancerogenesis [39]. This theory is supported by experimental data showing that targeted therapy reduced tumor cell proliferation via inhibition of tumor cell necrosis in GBM rat models [40,41].

This work shares the weaknesses of all retrospective studies. Selection bias cannot be fully ruled out. The investigated case number of mGBM seems relatively small but is comparable to previous studies, excluding the database analysis by Haque et al. Moreover, tumor necrosis volume failed to achieve statistical significance in our uGBM cohort. Reasons lie within selection criteria. All above-mentioned projects concerning the prognostic role of tumor necrosis volume in GBM included patients who received GTR only, defined a threshold of maximum residual tumor volume for eligibility or recruited in the pre-temozolomide era. Hence, the results presented here do not neglect previous findings (as it would not be possible due to their mono-centered, retrospective nature) but provide additional aspects to the clinical management of glioblastoma.

## 5. Conclusions

The prognosis of patients suffering from multifocal glioblastoma is very poor. As was shown before, craniotomy, employing maximum safe resection, is beneficial but already resecting more than 27.7% was associated with prolonged survival in univariate analysis for our cohort. The ratio of tumor necrosis volume to initial tumor volume seems to provide prognostic information for multifocal glioblastoma. In multivariate analysis, however, only adjuvant therapy regimen was associated with prolonged survival for patients suffering from mGBM.

**Supplementary Materials:** The following are available online at https://www.mdpi.com/article/10.3390/curroncol28020136/s1, Table S1: Cox regression with continuous, volumetric parameters, Table S2: Cutoff values calculated by ROC analysis, Table S3: Cox regression of categorized volumetric parameters, Table S4: Kaplan-Meier analysis of categorized, volumetric parameters, Table S5: Cox regression of continuous clinical and pathological parameters.

**Author Contributions:** Conceptualization, J.K.; data curation, N.H. and C.F.; formal analysis, J.K., N.H. and F.W.; methodology, T.W., M.K.F. and J.M.; project administration, J.M. and F.A.; software, G.H.; supervision, K.J., J.M. and F.A.; validation, J.M. and F.A.; writing—original draft, J.K.; writing—review and editing, J.K., N.H., T.W., M.K.F., F.W., K.J., C.F., G.H., J.M. and F.A. All authors have read and agreed to the published version of the manuscript.

**Funding:** We acknowledge support from Leipzig University for Open Access Publishing.

**Institutional Review Board Statement:** The study was conducted according to the guidelines of the Declaration of Helsinki and approved by the Institutional Review Board (or Ethics Committee) of Medical Faculty, University of Leipzig (No. 336/20-ek).

**Informed Consent Statement:** Informed consent was obtained from all subjects involved in the study.

**Data Availability Statement:** Data of this work is available from the corresponding author upon reasonable request.

**Acknowledgments:** We thank the Leipzig Cancer Registry for providing dates of death when not implemented in hospital archives.

**Conflicts of Interest:** The authors report no conflict of interest concerning the materials or methods used in this study or the findings specified in this paper.

## Abbreviations

AUC: area under the curve; CEV-: non-enhancing tumor area; CEV-/IC: ratio of CEV- to IV; CI: 95% confidence interval; EOR: extent of resection; GBM: IDH-wildtype glioblastoma; GTR: gross total resection; HR: hazard ratio; IV: initial tumor volume; RV: residual tumor volume; mGBM: multifocal GBM; MGMT: O6-methylguanine-DNA methyltransferase promoter methylation; MRC-NPS: medical research council neurological performance scale; MRI: magnetic resonance imaging; OS: overall survival; RCx: radio-chemotherapy; ROC: receiver operating characteristics; Rx: radiotherapy; SD: standard deviation; STR: sub-total resection; uGBM: unifocal GBM.

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
