# Peer review of "On the Prognosis of Multifocal Glioblastoma: An Evaluation Incorporating Volumetric MRI"

_curroncol, doi:10.3390/curroncol28020136_

Round 1

Reviewer 1 Report

The authors presents the results of the influence of radiological parameters on survival for unifocal versus multifocal GBM patients. The radiological parameters were derived form MRI. They report that the ratio of necrosis versus initial tumor volume and extent of resection were associated with prolonged survival in multifocal GBM; however, the statistical significance of such findings did not reach statistical significance on multivariate analysis.

The importance of MGMT gene promoter methylation status, age and adjuvant therapy remained statistically singnificant in multivariate analysis.

The additional value of the study is limited mainly on radiological parameters which are worth exploring further and integrating with clinical known prognostic factors. 

Author Response

The authors presents the results of the influence of radiological parameters on survival for unifocal versus multifocal GBM patients. The radiological parameters were derived form MRI. They report that the ratio of necrosis versus initial tumor volume and extent of resection were associated with prolonged survival in multifocal GBM; however, the statistical significance of such findings did not reach statistical significance on multivariate analysis.

The importance of MGMT gene promoter methylation status, age and adjuvant therapy remained statistically singnificant in multivariate analysis.

The additional value of the study is limited mainly on radiological parameters which are worth exploring further and integrating with clinical known prognostic factors. 

We thank the author for the positive feedback. Further studies are planned to explore the here presented data in detail.

Reviewer 2 Report

Difference between Figure 2 and 3 should be made more explicit. Furthermore, there are no explanations below both Figures explaining what is actually displayed.

Author Response

Difference between Figure 2 and 3 should be made more explicit. Furthermore, there are no explanations below both Figures explaining what is actually displayed.

We thank the reviewer fort the feedback. The data is presented in line 143-155 and further supported by Suppl. Table 3 and 4. Figure legends were edited to their corresponding figures. Small changes have been made to clarify the here presented data.   

Reviewer 3 Report

The article: volumetric assesment of multifocal glioblastoma and its prognostic factors, by J Kasper et al, is a well written article.

There some comment and questions, that could improve the information and quality of the article.

Introduction, line 48: combination temozolomide and lomustine has been reported in a study of selected patients with methylated MGMT, and was much criticized. I Believe it should not be mentionnedhere (irrelevant).

Line 53, initial tumor location (location in brain, not saem as uni or multifocal, I would only write unifocal or multifocal.

Methods, line 67: informed consent. This is a retrospective studye, did the authord get informed consent from patients (many of them would be dead when the study was done), by realtives, or is it a general consent that patient underwrite for example at the time of initial surgery ?

Line 70: exclusion. If patients did not get MRI within 72 hours from operation, have they been assessed by CT instead. How many patients have been excluded from the study cohort during the period.

Results, line 117: definition of multifocal: separate contrast enhancing in the same FLAIR region, or separate contrast enhancing in separate FLAIR (for example contralateral)

line 93 and 95: can authors explain concretely why they use residual volume for unifocal GBM, but the ratio between necrosis and initial volume in multifocal GBM. Especially related to data in table 1, where necrosis in mGBM is smaller than in uGBM. The comparison between residuel volume and necrosis/initial volume, for ex at line 137,  or supp table 2 is a Little confusing.

Figure 1: shows overall survival for the whole cohort and subcohort. But there is no data on the actual overall survival (how many months, median)

Fig 2F, it seems that the advantage or resection (27,7%), relies on the last 3 patients

Figure 3 D, E (F), is it possible to indicate the significance in the figure. Figure E : both curves are very Close, what is the significance?

Discussion line 207 (describes that necrosis is initiated by  depletion of energy, ie by hypoxia, but at line 211-212, it is necrosis that induces hypoxia, exactly yhe opposite. The discussion line 213-215 is irrelevant for this study.

Conclusion line 228, looking at figure  2F and 3 F  I would conclude that the time to progression is prolonged, the  I can estimate that  that the OS win is about 1 month after surgery, furthermore other factors  like: were the youngest patients that underwent most surgery, were the operated patients that got combined adjuvant therapy, were the smaller tumor with Little necrosis that underwent operation ? There are no data presented that could make such assumptions. The strenght of the article is on the prognosis of MRI, not on the extent of surgery.

Author Response

Introduction, line 48: combination temozolomide and lomustine has been reported in a study of selected patients with methylated MGMT, and was much criticized. I Believe it should not be mentionnedhere (irrelevant).

As sugggested, the citation was removed.

Line 53, initial tumor location (location in brain, not saem as uni or multifocal, I would only write unifocal or multifocal.

The line was changed accordingly.

Methods, line 67: informed consent. This is a retrospective studye, did the authord get informed consent from patients (many of them would be dead when the study was done), by realtives, or is it a general consent that patient underwrite for example at the time of initial surgery ?

All patients treated at our Neurosurgery Department consent for retrospective data analysis at admission. Therefore, further informed consent (i.e. by relatives) was declared not necessary by the local ethics commitee.

Line 70: exclusion. If patients did not get MRI within 72 hours from operation, have they been assessed by CT instead. How many patients have been excluded from the study cohort during the period.

The author is correct, patients who did not receive peri-operative MRI received CT imaging instead. Numbers concerning excluded patients was added within results (line 117).

Results, line 117: definition of multifocal: separate contrast enhancing in the same FLAIR region, or separate contrast enhancing in separate FLAIR (for example contralateral)

Multifocality was defined as separate (distance greater 1 cm) contrast-enhancing lesions, independently from FLAIR hyperintensity. The corresponding passage has been altered within methods.

line 93 and 95: can authors explain concretely why they use residual volume for unifocal GBM, but the ratio between necrosis and initial volume in multifocal GBM. Especially related to data in table 1, where necrosis in mGBM is smaller than in uGBM. The comparison between residuel volume and necrosis/initial volume, for ex at line 137,  or supp table 2 is a Little confusing.

Assessed radiological parameters were not compared between entities (i.e. between uGBM and mGBM) but rather within one entity. As our study aimed at exploring statistical significance of radiological parameters for multifocal GBM, only parameters with statistical significance (as shown in suppl. Table 1) were chosen for further analysis. In order to provide context, unifocal glioblastoma was added to this analysis. 

Figure 1: shows overall survival for the whole cohort and subcohort. But there is no data on the actual overall survival (how many months, median)

Figure 1 was created to illustrate the significantly worse prognosis of patients suffering from mGBM once more. 1-years-survival data derived from Kaplan-Meier analysis is provided within results (line 125 + 126). As we focused mainly on a more detailled survival analysis, further data exploring figure 1 was deemed not necessary.   

Fig 2F, it seems that the advantage or resection (27,7%), relies on the last 3 patients

Figure 2 represents curve data originated from Cox regression. Therefore, single patients data cannot be derived from it. Nevertheless the author is correct; due to a comparably smaller cohort (54 patients with mGBM) data of long-term survivors would have a larger impact than usual.

Figure 3 D, E (F), is it possible to indicate the significance in the figure. Figure E : both curves are very Close, what is the significance?

The corresponding statistical data for Kaplan Meier curves shown in figure 3 is presented in Suppl. Table 4. But the author anticipated correctly, the survival difference shown in 3 E is statistically not significant (p = 0.055) as also mentioned within results, line 151.

Discussion line 207 (describes that necrosis is initiated by  depletion of energy, ie by hypoxia, but at line 211-212, it is necrosis that induces hypoxia, exactly yhe opposite. The discussion line 213-215 is irrelevant for this study.

The author is correct, terms have been exchanged.

Conclusion line 228, looking at figure  2F and 3 F  I would conclude that the time to progression is prolonged, the  I can estimate that  that the OS win is about 1 month after surgery, furthermore other factors  like: were the youngest patients that underwent most surgery, were the operated patients that got combined adjuvant therapy, were the smaller tumor with Little necrosis that underwent operation ? There are no data presented that could make such assumptions. The strenght of the article is on the prognosis of MRI, not on the extent of surgery.

Time to progression was not assessed within this study and is hence not shown in Figures 2 and 3. The author is right, many clinical features, next to MRI-imaging data, influence patient overall survival. Knowing this and in order to rule out selection bias (i.e. through age, adjuvant therapy regimen etc.), we performed multivariate cox regression to evaluate statistical independency from corresponding factors (shown in table 2). Here, extent of resection was statistically not significant for patients with multifocal glioblastoma but was beneficial when employing univariate analysis. The corresponding sentence within the conclusions was clarified.